# Effect of Weaning at 21 Days of Age on the Content of Bile Acids in Chyme of Cecum

**DOI:** 10.3390/ani12162138

**Published:** 2022-08-20

**Authors:** Yu Zhang, Hongbing Xie, Lirong Wang, Jianhe Hu, Lei Wang, Shouping Zhang

**Affiliations:** 1College of Animal Science and Veterinary Medicine, Henan Institute of Science and Technology, Xinxiang 453003, China; 2Wuxi Application Technology Company Limited, Nantong 226200, China

**Keywords:** weaning piglets, cecum thyme, bile acids

## Abstract

**Simple Summary:**

Weaning increases the level of bile acids (BAs) in the cecum of piglets, and there is a certain interaction between group and weaning age on the level of BAs. However, studies on the effect of early weaning on intestinal BAs in piglets are not clear. Therefore, this experiment was conducted to investigate the changes of different BAs in the cecum chyme of weaned piglets, to reveal the relationship between weaning and changes of BAs in the intestine, and to provide a theoretical basis for early weaning and supplementary feeding of piglets. The results showed that weaning increased the content of BAs in the cecum of piglets, and there was an interaction between group and weaning age on the content of BAs.

**Abstract:**

This experiment was conducted to investigate the effects of weaning at 21 days of age on cecal chyme bile acids (BAs) in piglets. According to a 2 × 3 factorial design, the main factors were lactation and weaning, and the other factor was 22, 24, and 28 days of age, respectively. Piglets were randomly divided into two groups of eighteen piglets each and six piglets were selected for slaughter at 22, 24, and 28 days of age, respectively, to determine the content of different types of Bas in the intestinal lumen of the cecum. Results: (1) There was a significant interaction between weaning and age on intestinal primary Bas hyocholic acid (HCA) and chenodeoxycholic acid (CDCA) (*p* < 0.05), and weaning significantly increased the content of primary BAs in piglets’ intestines, which showed a trend of decreasing and then increasing with the increase in piglets’ age. (2) There was a significant interaction between weaning and age on intestinal secondary BAs deoxycholic acid (DCA), lithocholic acid (LCA), and ursodeoxycholic acid (UDCA) (*p* < 0.05). DCA and LCA in piglets’ intestines tended to decrease with increasing age, while UDCA showed a trend of decreasing and then increasing with increasing piglets’ age; weaning significantly increased the content of secondary BAs in piglets’ intestines. (3) There was a significant interaction between weaning and age on intestinal glycine chenodeoxycholic acid (GCDCA), taurochenodeoxycholic acid (TCDCA), and taurolithocholic acid (TLCA), but not on taurohyocholic acid (THCA), taurohyodeoxycholic acid (THDCA), and taurineursodeoxycholic acid (TUDCA) (*p* > 0.05). Weaning significantly increased the contents of GCDCA, TCDCA, TLCA, THDCA, and TUDCA in the intestinal tract (*p* < 0.05), while THCA content was not significant. In conclusion, weaning can increase the BAs content in the cecum of piglets, and there is an interaction between group and weaning age on BAs content.

## 1. Introduction

Modern intensive pig farming requires giving full play to the reproductive potential of sows, shortening the interval between litters, increasing the annual litter size of sows, and at the same time reducing the breeding cost of piglets; early weaning of piglets is one of the important measures to give full play to the reproductive potential of sows [1]. Early weaning of piglets can improve the utilization rate of sows, but the early loss of maternal antibody protection in piglets is not conducive to their development after weaning. Late weaning of piglets is better for piglet development, but the utilization rate of sows is greatly reduced, so finding a balanced weaning age for piglets is more critical. The weaning age of piglets is now shortened from the traditional 60 days to 28 days or even earlier in China [2]. However, early-weaned piglets have low immunity and an immature digestive system in the intestine, while piglets experience multiple factors such as environment, nutrition, and psychology, leading to lagging growth, reduced disease resistance, and even increased mortality [3,4]. For piglet weaning age, there are still some differences between large-scale pig farms and rural free-range farmers. Most large-scale pig farms wean at 21 days old, and some of them adopt weaning at 23 or 24 days old, while rural free-range farmers are weaning at 28 days old or 35 days old. In large-scale pig farms, especially in group farms, the weaning age is often used. In theory, the gastrointestinal tract of piglets at 21 days old can really live independently, and weaning at 21 days old can ensure the maximum utilization rate of sows. To know the health condition of weaned piglets, we analyzed the intestinal tract. The intestine is not only the largest digestive and absorption organ in animals, but also the largest immune organ in the body, and early weaning can affect intestinal health [5,6]. At present, there are more studies on the effect of weaning on piglets’ intestines, mainly focusing on the intestinal barrier, intestinal permeability, and the structure and diversity of microbiota in the intestine [7,8]. Bile acids (BAs), as metabolites of intestinal microorganisms, can promote nutrient absorption and play an important role in the regulation of intestinal physiology, and they are of great interest as participants in the metabolic pathway of the “hepatic-intestinal cycle” in animals [9,10]. BAs are divided into two categories, primary BAs and secondary BAs. Primary BAs are synthesized directly from cholesterol in the liver cells and are called primary BAs, including BAs and goose deoxycholic acid (GDCA). Secondary BAs are mainly formed in the intestinal tract after being broken down by bacteria and combined with sulfuric acid or glycine. However, studies on the effect of early weaning on intestinal BAs in piglets are still unclear. In this experiment, we investigated the changes of different BAs in the cecum chyme of weaned piglets to reveal the relationship between weaning and changes of BAs in the intestine, and to provide a theoretical basis for early weaning and supplementary feeding of piglets.

## 2. Materials and Methods

### 2.1. Experimental Animals and Experimental Design

Experimental piglets were raised in the Changping Base of State Key Laboratory of Animal Nutrition. This experiment used a 2 × 3 factorial randomized trial design with 2 factors for weaning and piglet age, with the main factor being lactation and weaning and the other factor being 22, 24, and 28 days of age, respectively. The selection of body conditions was similar to that of 6 nest large white, at 21 days of age. We selected 6 per litter head (6.1 + 0.2) kg weight of piglets were randomly divided into 2 groups, respectively for the weaning group (feeding based diet) and nursing group (in the room of origin continue breastfeeding and based diet food intake).

### 2.2. Experimental Diet and Feeding Management

In the experiment, all piglets were immunized and managed according to the routine management procedures of the pig farm. All pigs began to lure food from 7 days of age. Weaned piglets were fed in a single pen, the temperature and light time in the house were the same as those of lactating piglets, and the temperature was maintained at about 30 °C. During the experiment, the piglets drank water freely, the weaned piglets were fed with a corn-soybean meal diet at 08:00 and 16:00 every day. The weaned piglets’ diet was prepared into powder according to the nutrition standard of NRC (2012) of the United States, and the nutritional components met the nutritional requirements of 5–10 kg piglets. Table 1 shows diet composition and nutrient levels.

### 2.3. Sample Collection

At 22, 24, and 28 days of age, piglets were slaughtered in accordance with the principle of the same litter source and similar body weight. The piglets were euthanized with a sodium pentobarbital overdose of 40 mg/kg body weight, followed by exsanguination [11,12]. The piglets were quickly opened to the abdominal cavity for ligation and the cecum was separated, and the cecum contents were selected from the same position in the cecum into 2 mL cryopreservation tube with liquid nitrogen and then stored in the −80 °C refrigerator for testing.

### 2.4. Determination Index and Method

The BAs in intestinal contents were detected by water high-performance liquid chromatography-mass spectrometry. Except for hyocholic acid (HCA) and hyodeoxycholic acid (HDCA), standard products were purchased from the Steraloids company, other standard products were purchased from Sigma. This mainly included cholic acid (CA), chenodeoxycholic acid (CDCA), deoxycholic acid (DCA), lithocholic acid (LCA), ursolsdeoxycholic acid (UDCA), hyodeoxycholic acid (HDCA), hyocholic acid (HCA), glycine deoxycholic acid (GDCA), glycine chenodeoxycholic acid (GCDCA), glycoursodeoxycholic acid (GUDCA), glycine cholic acid (GCA), taurocholate acid (TCA), taurodeoxycholic acid (TDCA), taurineursodeoxycholic acid (TUDCA), taurolithocholic acid (TLCA), and taurochenodeoxycholic acid (TCDCA).

About 80 mg of lyophilized chyme was added into a 5 mL centrifuge tube. Add 500 μL 50 mm cold sodium acetate solution (pH = 5.6) to the centrifuge tube, then add 1.5 mL chromatographic grade methanol, vortex for 10 s, mix well, and incubate on a 40 °C 180 rmp shaking table for 1 h. Centrifuge at 4 °C for 20 min at 20,000 g, take 1.5 mL of supernatant into a 10 mL centrifuge tube, add 3 times the volume of sodium acetate for dilution, and vortex for 10 s to achieve the mixing effect. Through Bond Elute C18 solid phase chromatography column (500 mg/6 mL; Varian, Harbor City, CA, USA). First, wash with 5 mL of 25% chromatographic grade methanol, then elute with 5 mL of chromatographic grade methanol, and collect the eluent. Dry it with high-purity nitrogen, and the residue is redissolved in 1 ml of chromatographic grade methanol, and 0.22 μm filter membrane millipore filter (Millex^®^-lg; Billerica, MA, USA) to remove solid deposits.

### 2.5. Chromatographic and Mass Spectrometry Conditions

High-performance liquid chromatography (Thermo Dionex Ultimate 3000, Thermo Scientific, 81 Wyman Street, Waltham, MA, USA) was coupled with a Waters Xevo TQS triple quadrupole mass spectrometer (Waters, Milford, Mass, USA). The chromatographic column was Eclipse Plus C18 (2.1 × 100 mm, 1.7 μm). The mobile phases were: phase A (aqueous phase): 99.9% water, 1‰ formic acid; phase B (organic phase): 99.9% acetonitrile, and 1% formic acid. Flow rate: 0.3 mL/min; gradient elution program: 0–3 min, 67% A; 3–11 min, 67–10% A; 11–12.5 min, 10% A; 12.5–12.6 min, 10–67% A; 12.6–14 min, 67% A.

Triple quadrupole mass spectrometer Waters Xevo TQ-S (Waters, Milford, Mass, USA), ESI/APCI composite ion source, ion source temperature of 550 °C; positive ion mode scanning and MRM detection were used, where the scanning range was m/z 60–600, declustering, and collision voltage of 30–84 V and 3–84 V, respectively.

### 2.6. Data Statistics

Statistical ANOVA was carried out using the GLM in SPSS 22.0, and multiple comparisons were performed using Duncan’s method. All data were expressed as mean ± SEM.

## 3. Results

### 3.1. Effect of Weaning at 21 Days of Age on Primary BAs in Piglet Cecum Chime

As shown in Table 2, there was a significant interaction between group and age on the content of HCA in the cecum of piglets (*p* < 0.05), and the content of HCA in the cecum of the weaning group was significantly higher than that of the lactating group (*p* < 0.05). The increase in piglet age showed a trend of decreasing and then increasing, and the content of HCA in the cecum of piglets at 28 days of age was significantly higher than that of piglets at 22 and 24 days of age (*p* < 0.05), and the content of HCA in the intestine of piglets at 22 days of age was significantly higher than that of piglets at 24 days of age (*p* < 0.05). HCA content was significantly higher in piglets at 22 days of age than at 24 days of age (*p* < 0.05). There was a significant interaction between group and age on the content of CDCA in the cecum of piglets (*p* < 0.05), which was significantly higher in the weaning group than in the lactating group (*p* < 0.05), and significantly higher in piglets at 22 days of age than at 24 and 28 days of age (*p* < 0.05).

### 3.2. Effect of Weaning at 21 Days of Age on Secondary BAs in Piglet Cecum Chyme

As shown in Table 3, there was a significant interaction between group and day of age on the content of DCA in the cecum of piglets (*p* < 0.05), and the content of DCA in the cecum of the weaning group was significantly higher than that of the lactating group (*p* < 0.05), and that of piglets at 22 and 24 days of age was significantly higher than that of piglets at 28 days of age (*p* < 0.05). There was a significant interaction between group and age on the content of UDCA in the cecum of piglets (*p* < 0.05), and the content of UDCA in the cecum of the weaning group was significantly higher than that of the lactating group (*p* < 0.05), with a trend of decreasing and then increasing with the increase in piglet age, and the piglets at 22 days of age were significantly higher than those at 24 days of age (*p* < 0.05). There was a significant interaction between group and age on the content of LCA in the cecum of piglets (*p* < 0.05), which was significantly higher in the weaning group than in the lactating group (*p* < 0.05); this was significantly higher in piglets at 22 days of age than at 24 days of age, and significantly higher at 24 days of age than at 28 days of age (*p* < 0.05).

### 3.3. Effects of Weaning at 21 Days of Age on Glycine-Conjugated BAs of Cecum Chyme

As shown in Table 4, there was no significant interaction between group and day of age on the content of GUDCA in the cecum of piglets (*p* > 0.05). The content of GUDCA in the cecum of the weaning group was significantly higher than that of the lactating group (*p* < 0.05), and that of piglets at 28 days of age was significantly higher than that of piglets at 22 and 24 days of age (*p* < 0.05). There was a significant interaction between group and age on the content of GCDCA in the cecum of piglets (*p* < 0.05), and the content of GCDCA in the cecum of the weaning group was significantly higher than that of the lactating group (*p* < 0.05), which showed a trend of decreasing and then increasing with the increase in piglet age. The difference between 22 and 28 days old piglets was not significant (*p* > 0.05).

### 3.4. Effects of Weaning at 21 Days of Age on Taurine-Conjugated BAs of Cecum Chyme

As shown in Table 5, there was no significant interaction between group and age on the content of THCA in the cecum of piglets (*p* > 0.05), and there was no significant difference in the content of THCA in the cecum of the weaning group compared with the lactation group (*p* > 0.05). The intestinal THCA content of piglets was not significant (*p* > 0.05). There was no significant interaction between group and age on THDCA content in the cecum of piglets (*p* > 0.05), but THDCA content in the cecum of the weaning group was significantly higher than that of the lactating group (*p* < 0.05), and showed a trend of decreasing and then increasing with the increase in piglet age. The TUDCA content of piglets at 24 days of age was significantly lower than that at 22 days of age (*p* < 0.05). There was no significant interaction between group and age on the content of TUDCA in the cecum of piglets (*p* > 0.05), and the content of TUDCA in the cecum of the weaning group was significantly higher than that of the lactating group (*p* < 0.05), which showed a trend of decreasing and then increasing with the increase in piglet age. There was no significant difference (*p* > 0.05). There was a significant (*p* < 0.05) interaction between group and age on the content of TLCA in the cecum of piglets, which was significantly higher in the weaning group than in the lactating group (*p* < 0.05), and showed a tendency to decrease and then increase with increasing piglet age. There was no significant change in TLCA content between 24 and 28 days of age (*p* > 0.05). There was a significant interaction between group and age on the content of TCDCA in the cecum of piglets (*p* < 0.05), and the content of TCDCA in the cecum of the weaning group was significantly higher than that of the lactating group (*p* < 0.05), which showed a trend of decreasing and then increasing with the increase in piglet age. The content of TCDCA in the cecum of piglets at 28 days of age was significantly higher than that of piglets at 24 days of age (*p* > 0.05).

## 4. Discussion

BAs are a general term for a group of bile alkanoic acids that are metabolized from cholesterol in the hepatocytes and are the main components of bile [13,14]. The type and structure of BAs vary depending on the animal species, but in pigs, they are mainly α-deoxycholic acid, bilirubin, and goose deoxycholic acid. The composition structure of BAs plays a very important role in maintaining the physiology of intestinal homeostasis in piglets [15,16]. The intestine of piglets is almost sterile at birth, and 3–4 h after birth a large microbial community is formed in the intestine due to the influence of the maternal birth canal as well as the environment, and lactobacilli, bifidobacteria, *E. coli*, and enterococci can be detected in the intestine of piglets 24 h after birth. By the time the piglets are 3 days old, the intestinal flora is similar to that of the sow. When piglets are weaned, they not only suffer from the stress brought by separation from sows and environmental changes [17,18] but also suffer from the stress of changing from the original liquid breast milk to solid feed with poor palatability [19,20,21]. Therefore, it is very important to study the differences in the composition of BAs in the intestinal tract of piglets under weaning stress conditions.

BAs have a variety of physiological functions, due to their amphiphilic molecular structure, with an alkyl group at one end, which binds to lipids, and a carboxyl and hydroxyl group at the other end, which acts hydrophilically [22,23]. Therefore, BAs most notably facilitate the digestion and absorption of fatty acids and fat-soluble vitamins. According to their structure, they can be divided into two main groups, free BAs and conjugated BAs [24], which are the products of free BAs combined with glycine and taurine. BAs can be subdivided into primary and secondary BAs according to their sources; BAs synthesized directly from cholesterol in hepatocytes are primary BAs, and BAs formed after 7α-dehydroxylation after primary BAs enter the intestine are secondary BAs. Due to the wide variety of BAs, this test classified BAs into primary BAs: CA, CDCA; secondary BAs: DCA, LCA, UDCA, HDCA; glycosylated BAs: GUDCA, GCDCA; and taurocholic BAs: THCA, THDCA, TUDCA, TLCA, TCDCA [25].

Chen Chao et al. found that the amount and composition of BAs are intrinsically linked to the presence of microbial communities, perhaps to the composition of the communities, and that those specific combinations of microorganisms may lead to significant changes in BAs. Carlotta et al. showed that the microbiota in the intestine of animals is influenced by the diets they consume [26] and that different diets and ages can cause animals to respond to the nutrients they consume. Khan also found that the amount of solid feed consumed by piglets increased with increasing post-weaning age and that the increase in solid feed intake caused changes in the microbiota of the cecum [27]. In our experiments, we found that weaning significantly increased the content of primary BAs HCA and CDCA in the piglets’ intestines. The main reason for this is that on the one hand, the weaning of piglets on solid feed stimulates the secretion of BAs. On the other hand, due to the incomplete development of piglets’ intestinal tract, they have low immunity and unstable intestinal microflora. The intestinal environment is greatly affected by the change of nutrients from liquid milk to solid feed after weaning, and the physiological and psychological effects on piglets due to the change in the environment in which they live after weaning results in changes in the intestinal microflora, which in turn affects metabolism and reabsorption of Bas [28,29]. Impaired BAs metabolism can cause more severe intestinal inflammation [30,31]. Primary BAs are essential nutrients for the growth of *C. difficile* and can lead to the proliferation of *C. difficile*. A significant increase in the level of primary BAs at the beginning of weaning has the effect of promoting *C. difficile*, so it increases the incidence of diarrhea in piglets at the beginning of weaning, which is consistent with Xia Bing’s study [32]. Xia Bing found that the incidence of diarrhea in the weaning group was 4.63% compared to 0.93% in the lactating group [33,34]. Schokker et al. found that the intestinal flora of piglets undergoes drastic changes after weaning and that disruption of the intestinal microflora is the main factor leading to diarrhea in piglets [35].

Interestingly, the content of primary BAs HCA and CDCA and the content of secondary BAs UDCA in the cecum food morsels were found to decrease and then increase with the increase in piglets’ age in this experiment. This may be related to the intestinal environment of piglets, which was not established at the early stage of weaning, and the harmful bacteria such as E. coli were dominant in the intestinal tract, while the content of primary BAs in the intestinal tract increased significantly as the piglets adapted to solid feed and the intestinal environment was gradually stabilized [36]. Bacillus aerogenes, Bacillus commonus, Bifidobacterium, and Lactobacillus in the intestine can produce bile salt hydrolase to convert bound BAs into free primary BAs [37].

## 5. Conclusions

There was a significant interaction between group and age on the content of HCA, CDCA, UDCA, LCA, DCA, GCDCA, TLCA, and TCDCA in the piglets’ cecum (*p* < 0.05), and no significant interaction in the content of GUDCA, THCA, THDCA, and TUDCA (*p* > 0.05). Weaning increased the content of BAs in the cecum of piglets, and there was an interaction between group and weaning age on the content of BAs.

## Figures and Tables

**Table 1 animals-12-02138-t001:** Composition and nutrient levels of the basal diet (air-dry basis)%.

Ingredients	Content	Nutrient Levels	Content ^2)^
Corn	57.00	DE/(MJ/kg)	14.44
Soybean meal	20.00	CP	20.68
Full-fat soybean	6.00	Ca	0.70
Fish meal	5.00	TP	0.65
Whey powder	5.00	Lys	1.18
Limestone	0.51	Try	0.23
CaHPO4	0.50	Thr	0.89
Soybean oil	1.00	Met	0.37
Choline chloride	0.09	Met + Cys	1.23
Nacl	0.30		
Glucose	1.50		
Lys	0.40		
Met	0.10		
Thr	0.10		
Premix ^1)^	1.00		
Total	100.00		

^1)^ The premix provided the following per kilogram of diet; VA 18,000 IU, VD34500 IU, VE 22.5 mg, VK_3_ 4.5 mg, VB_1_ 4.32 mg, VB_2_ 12 mg, VB_6_ 4.86 mg, VB_12_ 0.03 mg, nicotinic acid 41.58 mg, pantothenic acid 33.12 mg, biotin 0.48 mg, folic acid 1.764 mg, Cu 20 mg, Fe 140 mg, Zn 140 mg, Mn 40 mg, Se 0.3 mg, I 0.5 mg. ^2)^ Nutrient levels were calculated values.

**Table 2 animals-12-02138-t002:** Effects of weaning at 21 days of age on primer BAs of cecum chyme.

Items	Days of Age/d	HCA (µmol/L)	CDCA (µmol/L)
Suckling group	22	127.62	10.25 ^d^
	24	72.66	1.51 ^e^
	28	144.63	30.18 ^b^
Weaning group	22	130.35	65.06 ^a^
	24	34.81	20.64 ^c^
	28	336.26	10.72 ^d^
Group			
Suckling group		114.97 ^b^	13.98 ^b^
Weaning group		167.14 ^a^	32.14 ^a^
Days of age			
22 days of age		128.99 ^b^	37.66
24 days of age		53.73 ^c^	11.08
28 days of age		240.44 ^a^	20.45
SEM		16.44	6.158
*p*-value			
Group		0.001	0.001
Days of age		<0.001	0.001
Group × days of age		<0.001	<0.001

^a, b, c, d, e:^ Means that do not share similar letter in row are significantly different, *p* < 0.05.

**Table 3 animals-12-02138-t003:** Effects of weaning at 21 days of age on secondary BAs of cecum chyme.

Items	Days of Age/d	DCA (µmol/L)	UDCA (µmol/L)	LCA (µmol/L)
Suckling group	22	4.77 ^c^	162.01	639.95 ^a^
	24	2.40 ^c^	57.73	174.39 ^c^
	28	2.80 ^c^	535.01	334.47 ^b^
Weaning group	22	45.75 ^a^	720.29	2023.13 ^a^
	24	45.89 ^a^	492.50	1100.09 ^b^
	28	14.95 ^b^	177.86	372.21 ^c^
Group				
Suckling group		3.32 ^b^	251.59 ^b^	382.94 ^a^
Weaning group		35.53 ^a^	463.55 ^b^	1165.14 ^b^
Days of age				
22 days of age		25.26 ^a^	441.15 ^a^	1331.54 ^a^
24 days of age		24.14 ^a^	275.12 ^b^	637.24 ^b^
28 days of age		8.87 ^b^	356.43 ^ab^	353.34 ^c^
SEM		4.386	46.63	99.23
*p*-value				
Group		<0.001	<0.001	<0.001
Days of age		0.001	0.005	<0.001
Group × days of age		0.002	<0.001	<0.009

^a, b, c:^ Means that do not share similar letter in row are significantly different, *p* < 0.05.

**Table 4 animals-12-02138-t004:** Effects of weaning at 21 days of age on Glycine-conjugated BAs of cecum chyme.

Items	Days of Age/d	GUDCA (µmol/L)	GCDCA (µmol/L)
Suckling group	22	1.86	0.64
	24	1.54	0.68
	28	3.63	1.82
Weaning group	22	9.97	5.39
	24	10.33	1.42
	28	19.63	4.18
Group			
Suckling group		2.34 ^b^	1.05 ^b^
Weaning group		13.3 ^a^	3.66 ^a^
Days of age			
22 days of age		5.92 ^b^	3.0 ^a^
24 days of age		5.93 ^b^	1.05 ^b^
28 days of age		11.63 ^a^	3.00 ^a^
SEM		2.41	0.587
*p*-value			
Group		<0.001	<0.001
Days of age		0.036	0.002
Group × days of age		0.210	0.007

^a, b:^ Means that do not share similar letter in row are significantly different, *p* < 0.05.

**Table 5 animals-12-02138-t005:** Effects of weaning at 21 days of age on taurine-conjugated BAs of cecum chyme.

Items	Days of Age/d	THCA (µmol/L)	THDCA (µmol/L)	TUDCA (µmol/L)	TLCA (µmol/L)	TCDCA (µmol/L)
Suckling group	22	1.40	2.77	2.72	0.26	1.00
	24	0.79	0.60	0.41	0.23	0.29
	28	0.25	7.23	7.86	0.98	0.57
Weaning group	22	1.04	4.14	5.30	8.39	5.21
	24	0.37	2.11	3.85	0.73	0.41
	28	0.43	11.27	11.67	1.05	4.22
Group						
Suckling group		0.812	3.53 ^b^	3.66 ^b^	0.492 ^b^	0.621 ^b^
Weaning group		0.614	5.84 ^a^	6.94 ^a^	3.39 ^a^	3.28 ^a^
Days of age						
22 days of age		1.219 ^a^	3.45 ^a^	4.01 ^b^	4.33 ^a^	3.11 ^a^
24 days of age		0.579 ^b^	1.352 ^c^	2.13 ^c^	0.48 ^c^	0.35 ^b^
28 days of age		0.341 ^b^	9.25 ^a^	9.77 ^a^	1.01 ^b^	2.40 ^a^
SEM		0.207	1.122	1.110	0.508	0.551
*p*-value						
Group		0.250	0.017	0.001	<0.001	<0.001
Days of age		0.001	<0.001	<0.001	<0.001	<0.001
Group × days of age		0.292	0.418	0.849	<0.001	0.002

^a, b, c:^ Means that do not share similar letter in row are significantly different, *p* < 0.05.

## Data Availability

The data sets during and analyzed during the current study are available from the corresponding author on reasonable request.

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
