# Peer review of "Effect of Weaning at 21 Days of Age on the Content of Bile Acids in Chyme of Cecum"

_animals, 2022, doi:10.3390/ani12162138_

Round 1
Reviewer 1 Report
The authors investigated the effects of weaning at 21 days of age on cecal chyme bile acids (BAs) in piglets. They also tried to relate the study results with production parameters of early weaning in pigs, such as low immunity, immature digestive system, disease resistance, and mortality. The subject of this study is suitable for the “Animal” journal. The authors suggest that early weaning increased the content of BAs in the cecum of piglets, and there was an interaction between the group and weaning age on the content of Bas. This is a fascinating study, and I suggest a few corrections to increase the scientific value of the manuscript. After these corrections are addressed by the authors, the manuscript can be accepted.
1. The authors should clearly explain why they chose the 21-day weaning age in the introduction and material method sections.
2. Bile acids examined in the study should be clearly stated in the aim of the study (lines 56-59).
3. The results have been well presented, and the results have been supported by the discussion, but the conclusion part should be arranged in a way that realistically emphasizes the results of the study.
Author Response
Dear reviewer,
Thank you for your comments on this article and your willingness to endorse our revised manuscript. The authors carefully revised the manuscript item by item according to your comments. All the authors would be very grateful if the revised manuscript could be approved by you.
Please see the attachment.
Thanks for your kind helps. I am looking forward to hearing from you.
Sincerely,
Assoc. Prof. Shouping ZHANG
College of Animal Science and Veterinary Medicine
Henan Institute of Science and Technology
Eastern of Hualan avenue, Xinxiang 453003, China
Tel: +86-0373-3040733
E-mail: zsp031659@126.com

Reviewer 2 Report
Investigating the effects of piglet weaning on the nutritional physiology of piglets is an important study for improving pig production.
This manuscript investigates the metabolism of bile acids, but there are many areas that are currently not worthy of publication, such as a lack of information.
There are no references that state that the weaning period was 60 days.
There is no mention of anesthetics or concentrations used to reduce animal distress.
As with all the resulting data, have you performed a normality test? Why did you choose the Duncan's test, which tends to show differences? In the absence of normality, you should choose a nonparametric method.
What are the units of the results? It is not stated in the table.
Because there is nothing related to diarrhea or other results of clinical observations, in the latter part of the discussion, the argument is too much up to the point, e.g., the part about c. difficle.
Author Response

(The authors gave the same response as above.)

Round 2
Reviewer 2 Report
If only procaine was administered for local anesthesia, it would be almost impossible to open the pig's abdomen. The dosage is also unknown.
It does not follow ethical guidelines for the use of animals in research and is difficult to publish.
Author Response
Manuscript ID: animals-1817119
August 10, 2022
Dear reviewer,
Thank you for your comments on this article and your willingness to endorse our revised manuscript. The authors carefully revised the manuscript item by item according to your comments. All the authors would be very grateful if the revised manuscript could be approved by you.
Please see the attachment.
Thanks for your kind helps. I am looking forward to hearing from you.
Sincerely,
Assoc. Prof. Shouping ZHANG
College of Animal Science and Veterinary Medicine
Henan Institute of Science and Technology
Eastern of Hualan avenue, Xinxiang 453003, China
Tel: +86-0373-3040733
E-mail: zsp031659@126.com
